# The Relation between Migratory Activity of *Pipistrellus* Bats at Sea and Weather Conditions Offers Possibilities to Reduce Offshore Wind Farm Effects

**DOI:** 10.3390/ani11123457

**Published:** 2021-12-04

**Authors:** Robin Brabant, Yves Laurent, Bob Jonge Poerink, Steven Degraer

**Affiliations:** 1Marine Ecology and Management (MARECO), Aquatic and Terrestrial Ecology (ATECO), Operational Directorate Natural Environment (OD Nature), Royal Belgian Institute of Natural Sciences (RBINS), Vautierstraat 29, 1000 Brussels, Belgium; ylaurent@naturalsciences.be (Y.L.); sdegraer@naturalsciences.be (S.D.); 2Ecosensys, Hoofdweg 46, 9966 VC Zuurdijk, The Netherlands; bob.jongepoerink@ecosensys.nl

**Keywords:** bats, bat migration, *Pipistrellus* sp., offshore wind farms, collision, mitigation, weather conditions

## Abstract

**Simple Summary:**

Some species of bats migrate over longer distances between their summer roosts and winter areas. During migration, many man-made obstacles, such as wind farms, can pose a collision risk for bats. As it is known that bats can fly over open sea during migration, and offshore wind farms can also be problematic. We studied the presence of bats during migration at several North Sea locations with the aim of understanding the weather conditions triggering bat migration at sea. Our results show a decrease in bat activity with distance from the coast and a correlation between bat migration and wind speed (negative), wind direction, temperature (positive), and atmospheric pressure (positive). Understanding these relationships can help in reducing the effects of offshore wind farms by periodically idling the blades when optimal meteorological conditions prevail and by opting for wind farm locations where bat activity is less prevalent.

**Abstract:**

Bats undertaking seasonal migration between summer roosts and wintering areas can cross large areas of open sea. Given the known impact of onshore wind turbines on bats, concerns were raised on whether offshore wind farms pose risks to bats. Better comprehension of the phenology and weather conditions of offshore bat migration are considered as research priorities for bat conservation and provide a scientific basis for mitigating the impact of offshore wind turbines on bats. This study investigated the weather conditions linked to the migratory activity of *Pipistrellus* bats at multiple near- and offshore locations in the Belgian part of the North Sea. We found a positive relationship between migratory activity and ambient temperature and atmospheric pressure and a negative relationship with wind speed. The activity was highest with a wind direction between NE and SE, which may favor offshore migration towards the UK. Further, we found a clear negative relationship between the number of detections and the distance from the coast. At the nearshore survey location, the number of detections was up to 24 times higher compared to the offshore locations. Our results can support mitigation strategies to reduce offshore wind farm effects on bats and offer guidance in the siting process of new offshore wind farms.

## 1. Introduction

Bats undertaking seasonal migration between summer roosts and wintering areas can cross large areas of open sea [1]. This is also the case in the southern North Sea, where bats have been frequently recorded in the last years (e.g., [2,3,4,5,6]). Nathusius’ pipistrelle (*Pipistrellus nathusii*) is the species that is most frequently reported at sea in Europe, but also common noctules (*Nyctalus noctula*), parti-colored bats (*Vespertilio murinus*), and Leisler’s bats (*Nyctalus leisleri*) have been observed [7,8,9].

Given the known impact of onshore wind farms on bats, offshore wind farms (OWFs) also pose risks to bats (e.g., [1,9,10,11,12,13]). Depending on the distance from the coast, OWFs can be out of the foraging range of local bats, but migratory bats may still be at risk. Sightings of roosting bats are regularly reported in OWFs (personal communication of OWF technicians), and surveys with acoustic bat detectors also confirmed the presence of bats in OWFs even at large distances from the coast [6,8]. These occurrences are generally limited to periods with calm weather, suitable for long-distance migration [14]. Hüppop and Hill [7] also hinted towards a weather dependency of the offshore presence of bats at a research platform 45 km off the German coast.

This study investigated the weather conditions linked to the occurrence of bats in the Belgian part of the North Sea (BPNS). We were particularly interested in shedding light on the weather conditions needed for offshore bat migration. A better comprehension of the phenology and weather conditions, which is also identified as a research priority by EUROBATS [1], would provide a scientific basis for mitigating the impact of wind turbines on bats at sea.

## 2. Materials and Methods

To study the presence of bats at sea, we installed ultrasonic detectors on offshore wind turbines and platforms for two autumn periods. The first survey was conducted in autumn 2017 in the C-Power offshore wind farm in the BPNS. The second survey, in autumn 2020, focused on three locations in the BPNS: the Norther wind farm, the Northwester II wind farm, and the measurement platform “Scheur-Wielingen”, which is part of the Flemish banks monitoring network of the Agency for Maritime Services and Coast. The wind farms are located between 23 and 49 km from the coast. The Scheur-Wielingen platform is a nearshore location, at 6 km from the coast (Figure 1).

The ultrasonic recorders (Batcorder 3.0/3.1 EcoObs Ltd., Nürnberg, Germany), registering the echolocation calls of bats, made full-spectrum recordings in the RAW format (sampling rate: 500 kHz; record quality: 20; threshold amplitude (sensitivity): −36 dB; post trigger: 400 ms in 2017, 200 ms in 2020; threshold frequency (sensitivity): 30 kHz in 2017, 16 kHz in 2020). Each recorder was powered by a solar panel. The recorded data were locally stored on SD memory cards. The detection range by a batcorder, for echolocation calls of the genus *Pipistrellus*, is between 17 and 35 m, when a threshold of −36 dB is set [15,16].

In 2017, we installed seven batcorders on seven different wind turbines in the C-Power wind farm (Figure 1). The batcorders were installed on the service platform of the turbines, at approximately 16 m above mean sea level (MSL).

The C-Power wind farm is located on the Thorntonbank in the BPNS at approximately 30 km from the nearest point of the Belgian coastline (Figure 1). The wind farm consists of 54 Senvion wind turbines and one offshore transformer platform. Six turbines have a capacity of 5 megawatts (MW), the other 48 are 6.15 MW turbines. The turbines have a cut-in wind speed (i.e., the minimum wind speed at which they operate) of 3.5 m/s, a rotor diameter of 126 m, and the hub height is approximately 94 m above MSL. The batcorders were installed on 8 August 2017 and were operational until 15 November 2017 (100 days).

A threshold frequency of 30 kHz was used to avoid wind turbine generated noise in the data set. This setting does not allow to reliably sample Nyctaloid bats (i.e., a species group that includes the genera *Nyctalus*, *Vespertilio*, and *Eptesicus*), which have a frequency of maximum energy (FME) lower than 30 kHz [17]. Therefore, this survey focused on pipistrelle bats only, which are most frequently recorded offshore.

In 2020, we installed three batcorders from 1 July until 15 November (138 days) on three platforms at sea: two transformer platforms in the OWFs Norther and Northwester II, at an altitude of, respectively, 19 m and 34 m MSL, and a third one on a measurement platform from the Flemish Agency for Maritime Services and Coast at 10 m MSL (Figure 1). The Norther wind farm, located at 23 km from the coast, has been fully operational since 2019 and consists of 44 Vestas V164-8.0 MW turbines. These turbines have a hub height of 107 m MSL and a rotor diameter of 164 m. The cut-in wind speed is 4.0 m/s. The hub height of the Northwester II turbines is also 107 m MSL, and the rotor diameter is 164 m but with a capacity of 9.5 MW with a cut-in wind speed of 3.0 m/s. That wind farm, at approximately 50 km off the coast, consists of 23 Vestas V164-9.5 turbines, operational since May 2020. The same batcorder settings were used as in 2017, except for the post trigger (200 ms) and threshold frequency (16 kHz). The threshold frequency was lowered compared to the 2017 surveys because experience showed that turbine generated noise did not prove to be problematic during surveys.

In this study, we only considered data from the migration season, August to mid-November, and focused on the *Pipistrellus* genus. Recordings of other species and recordings outside the migration season were not included in the analysis.

Detections were processed and visualized with the software program Sonochiro 3.3.3 (Biotope, France). Automated species identifications were verified by a bat expert. To level off high numbers of recordings caused by single individuals residing near a recorder, the recordings were converted to detection positive ten minutes (DP10), meaning that a ten-minute period was considered as positive if it contained at least one bat call (e.g., a specimen producing 100 calls in 10 min and a specimen only calling once were valued in the same way and rendered one DP10).

The meteorological variables wind speed, wind direction, atmospheric pressure, and air temperature were collected with a 10 min resolution at the C-Power offshore transformation platform (for the OWF locations) at an altitude of 38 m MSL and from the Scheur-Wielingen measuring platform, at 10 m MSL. Precipitation data were collected at the Zeebrugge port jetty (Figure 1). For further analysis, the meteorological values nearest to the time of a bat detection were linked to that detection. To account for the difference in occurrence of wind speed/wind direction during the surveys, total DP10 for a given wind speed/wind direction were divided by the occurrence (n) of the corresponding wind speed/wind direction. This is hereafter referred to as “adjusted DP10”. 

To assess the influence of wind speed and direction on the recorded bat activity, we calculated the tailwind component (TWC) and crosswind component (CWC). Calculations were performed following the methodology of Hüppop and Hilgerloh [18]:TWC = cos (observed wind direction − tailwind direction) × wind speed.
CWC = sin (observed wind direction − tailwind direction) × wind speed.

The tailwind direction is defined as the direction of migration minus 180°. Positive TWC values mean tailwind components, and negative values are headwind components. Crosswinds from the left of the bat are expressed by positive CWC values, and crosswinds from the right are expressed by negative CWC values.

Hüppop and Hill [7] assumed a WSW direction of bat migration in autumn for similar calculations with data from the FINO1 platform in the German Bight based on recoveries of ringed Nathusius’ pipistrelles [19,20]. Reproduction areas of Nathusius’ pipistrelle are located in Northeastern Europe, whereas the wintering areas are in the southwest [21]; this implies a SW migration during autumn. We calculated TWC and CWC for an assumed WSW direction of flight (i.e., bats crossing the North Sea towards the UK from the Dutch Zeeland coast, hereby passing the study site). Positive CWCs mean, in this case, offshore crosswinds.

We used a multiple regression model following the forward selection, as outlined in Zuur et al. [22], to identify environmental predictors of bat activity at sea. Detections were expressed as the number of DP10 per night. Both a Poisson distribution and negative binomial distribution proved unsuccessful, as there were too many zeros in the data set.

## 3. Results

All batcorders were operational without failure during 100 nights from 8 August until 15 November in 2017 and during 108 nights from 1 August until 15 November 2020. During both surveys, a total of 437 bat recordings, equaling 155 DP10, were made over 50 different nights (Table 1). Both in 2017 and 2020, bats were registered throughout the entire study period. In 2017, the number of detections peaked between the 23rd and 29th of September, when 54% of the recordings were made. In 2020, 93% of the recordings were made from 1 until 24 September (Figure 2). The recordings were identified as *Pipistrellus nathusii*, *P. pipistrellus*, or as *Pipistrellus* 40, a group name assembling *P. nathusii*, *P. pipistrellus*, and *P. kuhlii*.

The average number of DP10 per night at the nearshore location, Scheur-Wielingen, was 3.4–24.0 times higher than the number of DP10 per night at the offshore locations (Table 2). In Northwester II, the location that is the farthest distance from the coast, the average DP10/night was only 0.03 versus 0.67 at the nearshore location. These values indicate a decrease in bat presence with distance from the coast.

The recorded wind speed at night was, on average, 7.2 ± 4.1 m/s during the study period, with a median value of 6.6 m/s and a maximum of 27.4 m/s. The mean wind speed at the time bats were recorded was 3.4 ± 1.9 m/s, with a median value of 2.9 m/s, and 80.5% of the adjusted DP10 occurred when the wind speed was maximally 5 m/s (Figure 3). No bats were recorded when the wind speed was higher than 13.4 m/s.

The adjusted DP10 (i.e., DP10 per wind direction divided by the occurrence (n) of the corresponding wind direction) resulted in a clear peak in bat activity during easterly winds at the offshore survey locations (Figure 4A; Table 3). Nearshore, 29.6% of the adjusted DP10 were made when wind was coming from the east, 28.3% when wind was coming from the north, and 15.7% of adjusted DP10 were made during northeasterly winds. (Figure 4C; Table 3). During the 2017 and 2020 surveys, the wind was dominantly coming from the SW, both in the offshore and nearshore survey locations (Figure 4B,D).

On average, bats were detected during tailwind (mean TWC = 0.42 ± 3.52) and offshore crosswind conditions (mean CWC = 0.70 ± 2.29), when we assume that the direction of migration was WSW (Figure 5).

The mean atmospheric pressure at night during the study period was 1013.6 ± 8.4 hPa, with a median value of 1015.0 hPa. During bat recordings, the atmospheric pressure was 1018.9 ± 5.4 hPa with a median value of 1019.2 hPa. Nighttime temperature was 14.9 ± 3.3 °C on average during the study period. The mean temperature when bats were recorded was 16.0 ± 2.8 °C. Ninety percent of the recordings were made when temperature exceeded 13 °C and the atmospheric pressure was higher than 1015 hPa (Figure 6) or when wind speed exceeded 6 m/s (Figure 7). During the entire study period, it rained (i.e., >0.1 mm/10 min) 3.3% of the time. Only two DP10 occurred when it was raining. All other detections of bats were made during dry conditions.

## 4. Discussion

Bats were recorded from the end of August until mid-November in 2017 and from the beginning of August until the end of October in 2020. In both years, the number of recordings was highest in September with 75% and 93% of the recordings in 2017 and 2020, respectively. This coincides with the known migration peak period for Nathusius’ pipistrelle (i.e., mid-August to mid-October) [17,23]. Less detections than expected were recorded in August 2017 (study started 8 August), i.e., two DP10. As the weather in August 2017 was colder than average and with a lot of precipitation, we expect that the migration period of bats started later than on average.

The detections at the different survey locations showed that there was a negative relationship between the number of detections and distance from the coast. At the nearshore location, the number of detections was up to 24 times higher compared to the offshore locations. In general, the number of detections during these surveys was low compared with similar surveys on land during the autumn migration period. Three batcorders of the Lifewatch research infrastructure program located at the Belgian coast (Nieuwpoort, Oostende and the Zwin nature reserve) made 1193 records of *Pipistrellus* sp. bats from 1 August to 15 November 2020 [24]. In comparison, the three batcorders from our 2020 survey at sea only made 311 recordings. This suggests that only a small portion of the populations of Pipistrelle bats crossed the North Sea during the autumn migration. Acoustic surveys can, however, underestimate the actual activity of bats, as the detection range of the bat detectors is limited, typically between 17 and 35 m for *Pipistrellus* species, and varies with ambient conditions (temperature and humidity) and recorder settings [15,16]. In addition, bats may be less reliant on echolocation during their flight in an open marine environment and may also make use of visual cues as suggested by Eklöf et al. [25].

Our results corroborate the findings of previous studies that there is a positive relationship between the activity of migratory bats and ambient temperature and a negative relationship with wind speed [26,27]. The latter having a large influence on the presence of bats during our survey periods, with 80.5% of the DP10 recorded when wind speed was lower than 5 m/s. The relationship with atmospheric pressure is less clear. Our results suggest higher activity during higher pressure conditions, which was also found by Bender and Hartman [26] and Smith and McWilliams [28]. The opposite, increased activity during lower pressure conditions, has been observed in other studies (e.g., [10,29]).

Wind direction was deemed to be an important driver for autumn migratory bat activity at sea, with a clear peak in occurrence when wind originated from the E and, to a lesser extent, from the NE and SE. This coincides with the findings of Lagerveld et al. [14], who reported the highest bat activity at sea when the wind blew between the NE and SE. At the nearshore location Scheur-Wielingen, there was also a clear peak in detections (28.3%) during northerly wind, which was not the case at the other offshore locations. Possibly, these are individuals that followed the coastline in the SW direction and crossed open water from the Dutch Westkapelle to the Belgian coast, passing by the survey location Scheur-Wielingen (Figure 1).

By calculating the tailwind and crosswind components, the wind direction and wind speed at the time of bat recordings were combined. These results show, on average, slight tailwind and offshore crosswind conditions when assuming a WSW migration direction. Hüppop and Hill [7] reported that the highest bat activity at a research platform in the German Bight coincided with offshore crosswind conditions, suggesting that wind drift is the main driver of bat occurrence at sea. Given the high number of recordings during low wind speed conditions, it seems unlikely that offshore drift fully explains the occurrence of bats in our study sites. It might be the case for some recordings, but intentional migration across the North Sea channel seems more likely for most offshore recordings, and these results suggest that they make use of favorable wind conditions doing so.

During migratory activity over sea, bats can be negatively affected by OWFs. The negative impacts of wind farms on land on bats are elaborately described in multiple studies (e.g., [1,10,11,12,13]), where the mortality of bats is increased by collision with turbines and barotrauma [30,31,32]. Given the known impact of wind turbines on land, offshore wind farms could also be a problem for bats [23,33]. Although Ahlén et al. [34] expect that accidents with wind turbines are probably not frequent during migration itself, because bats generally migrate at low altitudes, other studies at sea showed that bats also migrate at higher altitudes (>100 m) [35] and at nacelle height [6].

Our results can support mitigation strategies to reduce OWF effects on bats. The relation between migratory bat activity and ambient temperature and wind speed offer possibilities to do so. Wind turbines become operational at a certain minimum wind speed, i.e., the cut-in wind speed. For wind turbines in the North Sea, this is generally between 3 and 4 m/s. As shown in these surveys, respectively, 61.1% and 70.7% of the bat activity occurred when wind speed was lower than these values. Increasing the cut-in wind speed to 5 m/s, would allow for no less than 80.5% of the bat activity to take place when turbines were inactive. Therefore, this can be considered an effective mitigation measure, which is also supported by other studies [36] and is already being imposed in the Dutch Borssele wind farms between 15 August and 31 October [37]. In 2020, there was also a remarkable number of detections during the night of 12 and 13 July at the three survey locations at sea (unpublished data). During that night, wind was blowing at 5 m/s from the ENE. The beginning of July is known as the period when young individuals fly for the first time from their roost [21]. These recordings at sea are thus possibly linked to dispersing young individuals. This shows that certain individuals also fly over the North Sea outside of the known migration period and that extending the increase in cut-in wind speed should be considered also outside the known migration period. On the other hand, the measure could be abolished when ambient temperature at night is lower than 13 °C, when only 10% of the migratory bat activity was recorded.

The decrease in the number of detections with distance from the coast offers a tool in the siting process of OWFs. It can be an argument not to build wind farms in coastal and nearshore areas and outside of the foraging range of local bats. However, even when sited further offshore, OWFs will still pose a risk for migratory bats crossing the North Sea. Mitigation strategies, based on results of this study and other studies, can reduce the impact on bats.

## 5. Conclusions

EUROBATS identified studies that investigate if there is offshore bat activity and under which weather conditions this takes place are a research priority [1]. Studies such as ours help to address these questions and offer tools to reduce OWF effects on bats. A flexible measure where turbine cut-in wind speed can be adjusted based on wind speed and ambient nighttime temperature can reduce the number of fatalities at a minimal economic cost. The negative relationship between offshore bat activity and distance to shore is an argument to avoid building OWFs in coastal and nearshore areas.

## Figures and Tables

**Figure 1 animals-11-03457-f001:**
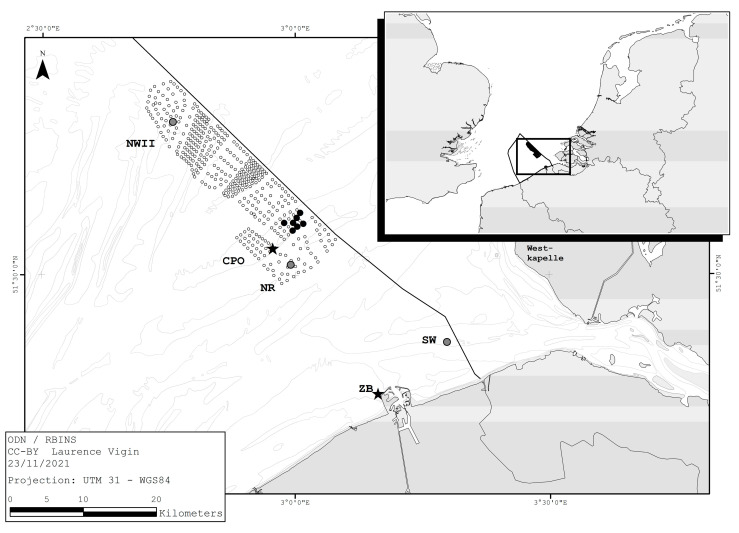
Location of the batcorders during the 2017 and 2020 surveys in the Belgian part of the North Sea. In 2017, seven batcorders were installed on different wind turbines in the in the northeast of the C-Power (CPO) wind farm on the Thorntonbank (black dots). In 2020, three batcorders were installed on the offshore platforms of the Norther (NR) and Northwester II (NWII) wind farms and on the Scheur-Wielingen (SW) platform (grey dots). Meteorological data were collected at the C-Power offshore transformation station (star), Scheur-Wielingen, and Zeebrugge (ZB) port meteopark (star). Wind turbines are shown as small dots.

**Figure 2 animals-11-03457-f002:**
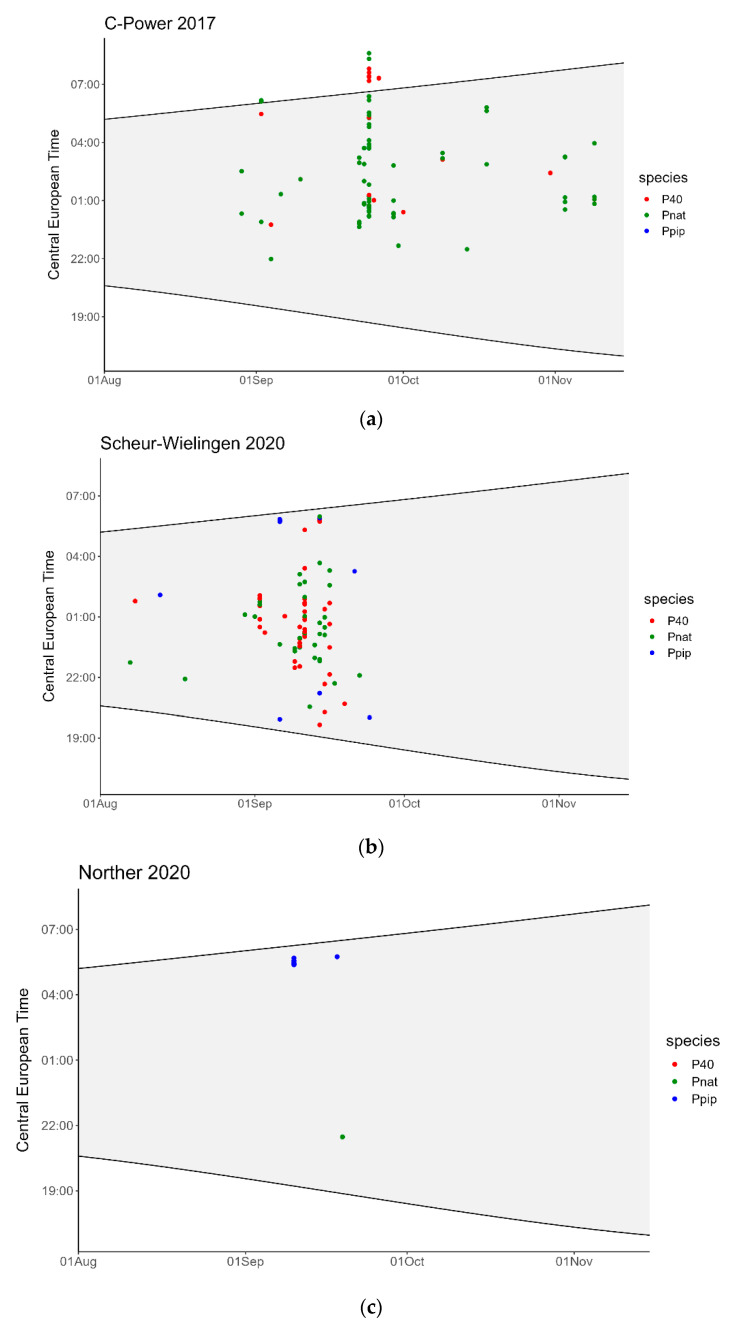
Actograms of all recordings of bat call sequences during the 2017 and 2020 surveys for the different survey locations: (**a**) C-Power; (**b**) Scheur-Wielingen; (**c**) Norther; (**d**) Northwester II. The C-Power graph includes the data for all seven batcorders that were used in that survey. Sunrise and sunset are indicated by the black lines. Pnat: *Pipistrellus nathusii*; Ppip: *P. pipistrellus*; P40: *P. nathusii*/*P. pipistrellus*/*P. kuhlii*.

**Figure 3 animals-11-03457-f003:**
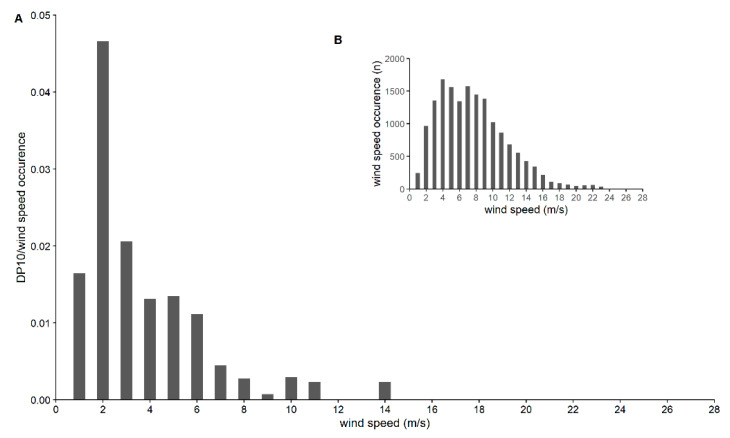
Bat activity in relation to wind speed, expressed as detection positive 10 min (DP 10) for a given wind speed divided by the occurrence (n) of the corresponding wind speed (**A**). Wind speed occurrence (n) measured during the survey periods (**B**). The range on the x-axis is the range of wind speed measured during the study period.

**Figure 4 animals-11-03457-f004:**
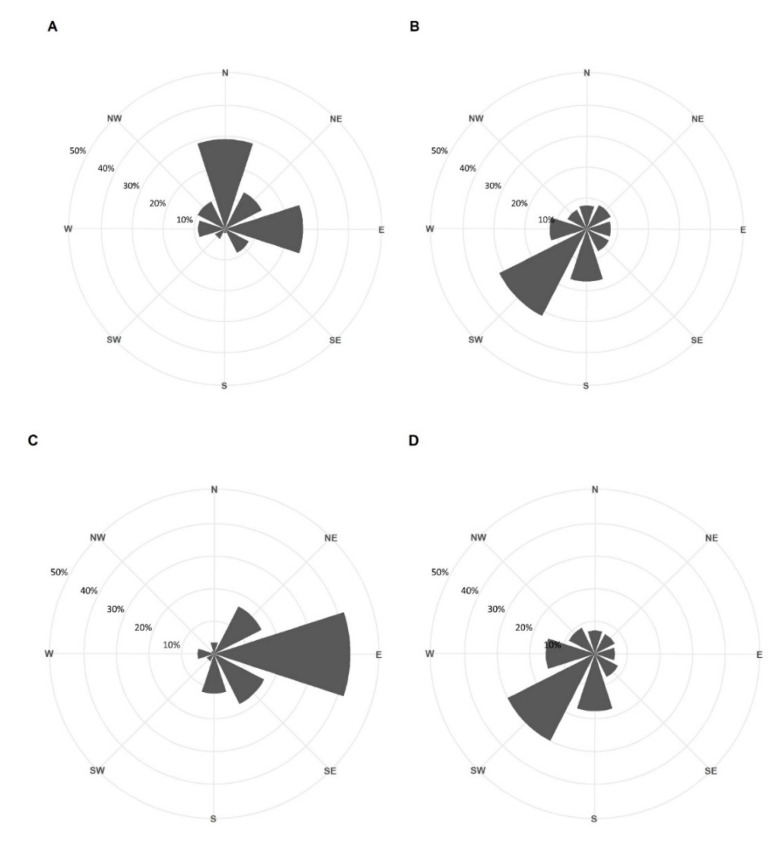
Percentage of adjusted bat activity in relation to wind direction (i.e., detection positive ten minutes (DP 10) of bat recordings for a given wind direction, divided by the frequency of occurrence (n) of the corresponding wind direction during the study periods) (**A**) for the nearshore location (Scheur-Wielingen) and (**C**) for the offshore survey locations (Norther, C-Power, and Northwester II). Percentage of wind direction occurrence during the 2017 and 2020 surveys at (**B**) the nearshore location and (**D**) the offshore locations.

**Figure 5 animals-11-03457-f005:**
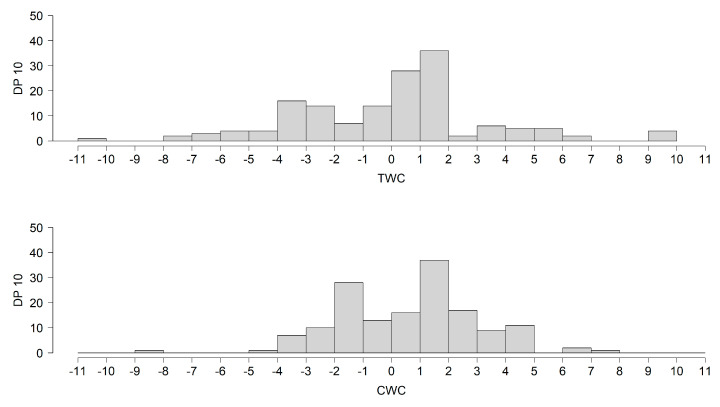
Detection positive ten minutes (DP10) of bat recordings in relation to the tailwind and crosswind components, assuming a WSW migration direction. Positive TWC = tailwinds; negative TWC = headwinds. Positive CWC = offshore wind; negative CWC = onshore wind.

**Figure 6 animals-11-03457-f006:**
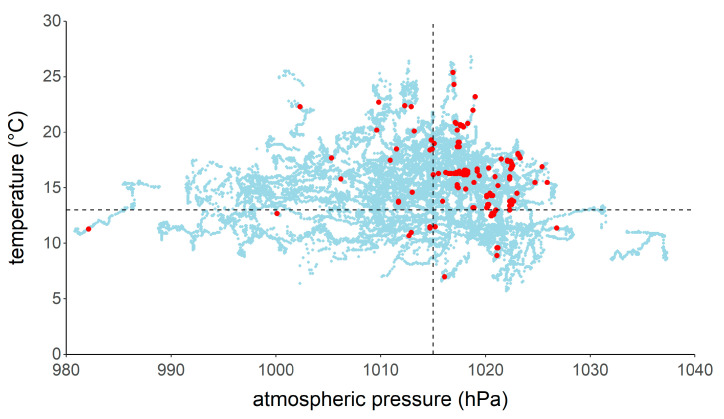
Temperature and atmospheric pressure measurements during the survey periods (light blue dots) and temperature and atmospheric pressure values when bats were recorded (red dots). Ninety percent of bat recordings were made when the temperature exceeded 13 °C (horizontal dashed line) and atmospheric pressure was higher than 1015 hPa (vertical dashed line).

**Figure 7 animals-11-03457-f007:**
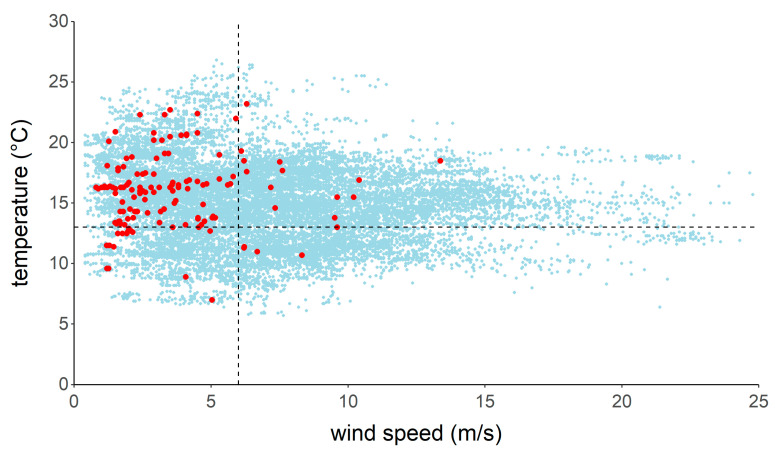
Temperature and wind speed measurements during the survey periods (light blue dots) and temperature and wind speed values when bats were recorded (red dots). Ninety percent of bat recordings were made when the temperature exceeded 13 °C (horizontal dashed line) and wind speed was lower than 6 m/s (vertical dashed line).

**Table 1 animals-11-03457-t001:** Number of records and detection positive ten minutes (DP10) per species for each survey location. Each location in the C-Power wind farm represents a turbine with its unique name. Pnat: *Pipistrellus nathusii*; Ppip: *P. pipistrellus*; P40: *P. nathusii*/*P. pipistrellus*/*P. kuhlii*.

Year	Location	*N* Records	DP10	Species
2017	C-Power G01	19	9	Pnat
		6	3	P40
	C-Power G03	30	9	Pnat
		5	3	P40
	C-Power H01	7	3	Pnat
		4	1	P40
	C-Power H02	25	18	Pnat
		7	2	P40
	C-Power I01	17	9	Pnat
		1	1	P40
	C-Power I03	7	4	Pnat
	C-Power J01	16	8	Pnat
		7	5	P40
2020	Norther	11	4	Ppip
		1	1	Pnat
	Northwester II	1	1	Ppip
		2	1	Pnat
		1	1	P40
	Scheur-Wielingen	31	7	Ppip
		111	32	Pnat
		128	33	P40

**Table 2 animals-11-03457-t002:** Detection positive ten minutes (DP10) per night, per acoustic detector during the 2017 and 2020 surveys. Each location in the C-Power wind farm represents a turbine with its unique name.

Year	Location	Distance from the Coast (km)	DP10/Night
2017	C-Power G01	29.3	0.12
2017	C-Power G03	30.6	0.12
2017	C-Power H01	29.5	0.04
2017	C-Power H02	30.4	0.20
2017	C-Power I01	29.4	0.10
2017	C-Power I03	30.8	0.04
2017	C-Power J01	31.2	0.13
2020	Norther	23.4	0.05
2020	Northwester II	48.7	0.03
2020	Scheur-Wielingen	6.4	0.67

**Table 3 animals-11-03457-t003:** DP10 per wind direction divided by the occurrence (n) of the corresponding wind direction (i.e., adjusted DP10) presented as percentages for the offshore and nearshore survey locations.

Wind Direction	Adjusted DP10 (%) Offshore	Adjusted DP10 (%) Nearshore
E	41.9	29.6
N	3.6	28.3
NE	16.4	15.7
NW	1.0	7.2
S	12.2	1.2
SE	17.3	8.1
SW	2.6	3.6
W	5.1	6.4

## Data Availability

The meteorological data and bat survey data supporting the reported results can be found in the Appendix A to this paper. The acoustic recordings are available upon request.

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
