# Peer review of "The Relation between Migratory Activity of Pipistrellus Bats at Sea and Weather Conditions Offers Possibilities to Reduce Offshore Wind Farm Effects"

_animals, 2021, doi:10.3390/ani11123457_

Round 1
Reviewer 1 Report
The paper is clearly written and very important data and conclusions for the conservation of offshore migrating bats. It has to be published absolutely!

Reviewer 2 Report
This paper is nearly without fault and presents some clear findings that do seem to have practical application in determining, as the authors discuss, conditions under which to curtail and locations in which wind farm development might have minimized impact on migrating bats.
I have one very significant complaint. I've never seen a peer-reviewed submission that totally lacked statistical tests before. This may be appropriate for the journal, and so I defer to the managing editor to weigh the importance of this complaint. In most cases, sufficient information is given to assess that the conditions of, for instance, temperature during which bat activity was high readily appeared to be a non-random sample of the range of conditions at the sites. For instance, the pattern of activity with temperature should be compared to temperatures during the night during the migratory months. Maybe it already is. For wind speed and direction, the prevailing conditions are not described at all. I would like to see the full range of conditions relevant to the times bats were detected and some assessment of whether their behavior is non-random.
Specific comments
57: scientific basis
162: X times lower is an unusual way to present a comparison. Could you reverse the statement (as in the figure caption) to fit the format "X times higher"?
164: It may help the reader to give context to 0.03. "versus 0.67 near shore."
202: Normalization by detection is insufficient to conclude that the pattern is selective. Need a comparison with the distribution of wind directions.
208: Figure 4 is so attractive I almost forgot to ask what the different colors indicate. Just different directions? It would be helpful if the full distribution of wind directions was presented here, as well (over nights during months with activity).
223: What % of nights exceeded 13 degrees?
225: Delete "during"
228: Could Figures 6 and 7 include separate dotted lines indicating mean conditions when bats were detected?
280: It would be easier for the reader if you use words rather than abbreviations here.
309: Replace "most likely" with "possibly". Please explain more fully or give references for this inference.
Reviewer 3 Report
P1, line 18: clarify that this will reduce the *negative impact of wind farms, not just general effects
P1, line 20: "areas" twice in one sentence, consider using another word
P2, line 52: *off* the German coast
P2, line 60: which detectors did you use? I recommend mentioning it here when you first bring them up. was there only one nearshore location and 3 offshore locations?
P2, line 67: delete the . in (Figure. 1)
P2, line 82: if the detectors were installed 16 m above msl, but the hubs (where the bats are hitting) are 94 m above msl, it is worth mentioning the general range of the microphones to acknowledge whether you could potentially be missing calls from the impact zone
P2, line 89: I don’t think you need to remind the reader to see above. We know from the intro that pips are the most often recorded at sea.
P2, line 96: are there extra spaces at the beginning of the “These turbines…” sentence?
P3, line 99: *off the coast. “… which are operational since May 2020.” Is a bit weird to read, consider rewording
P3, line 100: *settings
P3, lines 100-103: citation?
P3, line 108: all of the calls or just a portion were verified?
P3, line 118: Change “fig. 1” to “Figure 1” to match the others
P3, lines 118-119: “For further analysis…” This sentence is confusing. Didn’t you collect the variables at 10-min resolution? So they should have matched up?
P3, line 122: Choose whether windspeed is one word or two and stick with it throughout the text. Do not need - before direction
P4, Figure 1: I do not see Scheur-Wielingen and Zeebrugge (ZB) port metropark on the map. Am I missing it? Are all of the detectors just in the one spot for each site?
P4, lines 147-151: If both poisson and nb were unsuccessful, then what did you do? What kind of multiple regression model?
P4, line 157: When you mention other dates you do not put the superscript letters next to the numbers (e.g. 23rd and 29th). Delete this to be consistent with the rest of the text (same thing later on in the paragraph).
P4, line 160: lowercase pipistrellus for P. pipistrellus
P4, line 164: capitalize Table 2
P5, Tables 1 and 2: capitalize column titles
P8, line 202: Be consistent throughout whether or not north-easterly is capitalized or not.
P8, Table 3/Figure 4: Only pip bats, right? Figure 4 is really cool
P9, line 223: I don’t think you can start a sentence with a number, need to write out ninety percent (same for Figure 6 and 7 captions).
P9, line 225: When you originally bring up precip in the methods, would be good to define how much precip you considered to be “raining” (e.g., >0.01mm/10 mins)?
P10, line 245: Here and throughout you switch between e.g. and i.e. Choose one. Same goes for referring to wind directions in words (northeasterly) vs just abbreviations (NE)
P10, line 246: Change 8th to 8.
P10, paragraph 2 of Discussion: this entire paragraph is just restating results with a tiny bit of discussion at the end. It would be nice to have this developed further.
P11, lines 290-297: This is intro, not discussion
P11, lines 307-309: This is interesting and should also be mentioned in the results if you are bringing it up as a discussion point
P11, lines 315-319: These sentences are a bit awkward, consider rewriting within the context of the discussion, esp since foraging bats are not the focus of the study.
P12, line 321: Why is this first sentence here?
